# Oxidative Stress and Cerebral Vascular Tone: The Role of Reactive Oxygen and Nitrogen Species

**DOI:** 10.3390/ijms25053007

**Published:** 2024-03-05

**Authors:** Michele Salvagno, Elda Diletta Sterchele, Mario Zaccarelli, Simona Mrakic-Sposta, Ian James Welsby, Costantino Balestra, Fabio Silvio Taccone

**Affiliations:** 1Department of Intensive Care, Hôpital Universitaire de Bruxelles (HUB), 1000 Brussels, Belgium; 2Institute of Clinical Physiology—National Research Council (CNR-IFC), 20133 Milan, Italy; 3Department of Anesthesiology, Duke University Medical Center, Durham, NC 27710, USA; 4Environmental, Occupational, Aging (Integrative) Physiology Laboratory, Haute Ecole Bruxelles-Brabant (HE2B), 1160 Brussels, Belgium; 5Anatomical Research and Clinical Studies, Vrije Universiteit Brussels (VUB), 1050 Elsene, Belgium; 6DAN Europe Research Division (Roseto-Brussels), 1160 Brussels, Belgium; 7Motor Sciences Department, Physical Activity Teaching Unit, Université Libre de Bruxelles (ULB), 1050 Brussels, Belgium

**Keywords:** oxidants, reactive oxygen species, reactive nitrogen species, antioxidants, cerebral vascular tone

## Abstract

The brain’s unique characteristics make it exceptionally susceptible to oxidative stress, which arises from an imbalance between reactive oxygen species (ROS) production, reactive nitrogen species (RNS) production, and antioxidant defense mechanisms. This review explores the factors contributing to the brain’s vascular tone’s vulnerability in the presence of oxidative damage, which can be of clinical interest in critically ill patients or those presenting acute brain injuries. The brain’s high metabolic rate and inefficient electron transport chain in mitochondria lead to significant ROS generation. Moreover, non-replicating neuronal cells and low repair capacity increase susceptibility to oxidative insult. ROS can influence cerebral vascular tone and permeability, potentially impacting cerebral autoregulation. Different ROS species, including superoxide and hydrogen peroxide, exhibit vasodilatory or vasoconstrictive effects on cerebral blood vessels. RNS, particularly NO and peroxynitrite, also exert vasoactive effects. This review further investigates the neuroprotective effects of antioxidants, including superoxide dismutase (SOD), vitamin C, vitamin E, and the glutathione redox system. Various studies suggest that these antioxidants could be used as adjunct therapies to protect the cerebral vascular tone under conditions of high oxidative stress. Nevertheless, more extensive research is required to comprehensively grasp the relationship between oxidative stress and cerebrovascular tone, and explore the potential benefits of antioxidants as adjunctive therapies in critical illnesses and acute brain injuries.

## 1. Introduction

Brain tissue is exceptionally susceptible to the harmful effects of oxidative stress due to its unique characteristics. Despite accounting for only 2% of the body weight, the brain consumes approximately 20% of the total basal oxygen [1,2]. This heightened oxygen consumption is essential for ATP production, which is necessary to maintain the brain’s high metabolic rate. Unfortunately, the electron transport chain in the mitochondria, responsible for ATP synthesis, leads to the leakage of reactive oxygen species (ROS) [3]. Consequently, the brain cells ceaselessly generate a significant amount of ROS. Additionally, the brain comprises non-replicating neuronal cells, possesses a high cell surface-to-cytoplasm ratio, and has a weak antioxidant capacity and a relatively low repair capacity. These structural and functional features further contribute to its high susceptibility and vulnerability to oxidative damage [4].

The regulation of vascular tone in cerebral vessels is controlled by both muscular and endothelial mechanisms [5]. The former is regulated by myogenic mechanisms originating from vascular smooth muscle. The latter is regulated by endothelial factors, such as nitric oxide and endothelin, which can decrease or increase the vascular tone, respectively. In addition, local hormones, metabolic by-products, ROS, reactive nitrogen species (RNS), and medications can influence and contribute to cerebral vascular tone. Notably, ROS can arise from all the layers of the vascular wall and adjacent tissues through several mechanisms, and, among these, the NADPH-oxidase enzyme may be the primary source of ROS production from the vascular cells [6].

This widespread existence of ROS within and surrounding cerebral blood vessels aligns with their ability to influence the vascular tone and vascular permeability [7,8,9,10], and their effects may be greater in the cerebral vasculature than in any other areas [11,12]. Even cerebral autoregulation (CAR), a complex physiological mechanism that adjusts cerebrovascular tone to maintain cerebral blood flow within desired ranges in response to changes in cerebral perfusion pressure, may be altered by ROS [13]. All these vascular effects may primarily come from the direct activity of ROS on ion channels [14,15], through their interrelation with RNS, particularly nitric oxide (NO), or other unknown vasoactive mechanisms. The balance of redox reactions, given by ROS and RNS, may be of clinical interest, especially in critically ill settings when high oxygen levels are frequently given to the patients (thus increasing reactive species production), or in acute brain injuries when these species are more frequently produced [16,17].

In the last few years, there has been an increasing interest in the physiology of ROS and RNS and how they can impact a patient’s clinical status. Similarly, antioxidants have been applied as an adjunctive therapy in several clinical conditions. A thorough understanding of the mechanisms that regulate cerebrovascular tone is essential for further improving our knowledge of various brain injuries’ pathophysiology. This narrative review offers a comprehensive understanding of ROS and RNS, and delivering a contemporary summary of research on how oxidants and antioxidants affect cerebrovascular tone modulation. It facilitates more knowledgeable investigations into this field, aiming to improve therapies for brain injury patients, for whom slight alterations in vascular tone may result in worsened outcomes.

## 2. Experimental Approach and Methodology

A comprehensive search was performed to gather relevant literature from PubMed, Scopus, and Google Scholar databases. The search covered studies published up to August 2023, utilizing combinations of keywords including “oxidative stress”, “reactive oxygen species”, “reactive nitrogen species”, “cerebral vascular tone”, and “antioxidants”. Both text words and, where applicable, medical subject headings (MeSH) were employed to ensure a broad retrieval of pertinent studies. The search was limited to articles published in English. The selection criteria were designed to include studies contributing to understanding the interplay between oxidative stress and cerebral vascular tone, including original research articles, comprehensive review articles, and meta-analyses. The articles’ references were also reviewed for additional papers to further explore relevant aspects.

## 3. Cerebral Vascular Tone

The vascular tone is defined by the contractile state of the vascular smooth muscle cells lining the blood vessels [18]. The cerebral vascular tone contributes to modulating cerebral blood flow (CBF) [19], ensuring sufficient oxygen and nutrients reach the brain tissue. In physiological situations where intracranial hypertension is absent, cerebrovascular tone predominantly defines the effective downstream pressure in the cerebral circulation, highlighting its significance as the principal factor in regulating this pressure [20]. For this reason, the maintenance of cerebrovascular tone plays a key role in the proper functioning of the cerebral vasculature, contributing to balancing the needs of the brain’s metabolic demands with the systemic blood pressure and preventing conditions such as ischemia or hemorrhage, which can occur when CBF is either too low or too high, respectively. Disruptions in cerebrovascular tone regulation can contribute to insults in cerebrovascular diseases [21], such as stroke or brain injury.

The regulation of cerebrovascular tone involves neural, endothelial, and myogenic mechanisms [5]. Neural control is mediated primarily through the autonomic nervous system, with sympathetic, parasympathetic, and non-adrenergic non-cholinergic (NANC) pathways modulating vessel diameter. Myogenic mechanisms respond to changes in intravascular pressure to maintain constant CBF during fluctuations in systemic blood pressure, a phenomenon known as autoregulation. The endothelium contributes to cerebrovascular tone by producing vasodilators such as nitric oxide (NO) and prostacyclin, and vasoconstrictors like endothelin 1. Furthermore, cerebrovascular tone is influenced by metabolic factors, including CO_2_ and O_2_ levels [22,23]. Hypercapnia leads to vasodilation, increasing CBF, while hypocapnia causes vasoconstriction, reducing CBF. Similarly, hypoxia can induce vasodilation to enhance blood flow to oxygen-deprived areas of the brain [24]. Therefore, the regulation of cerebrovascular tone involves a complex interaction of multiple factors, including oxidative status, and is vital for preserving cerebral homeostasis, as its disruption can result in neurological deficits. To investigate the effects of oxidative stress in cerebral vasculature, a variety of experimental models and techniques have been explored in the literature and research, as outlined also in this review. These methods are summarized in Table 1.

## 4. Reactive Oxygen Species

The term Reactive Oxygen Species (ROS) refers to oxygen-containing chemical species that are unstable and highly reactive, capable of causing damage to various biological molecules. These species arise from the uptake of electrons from molecular oxygen, O_2_.

The main mechanisms of the formation of ROS are depicted in Figure 1. Several mechanisms contribute to the generation of reactive oxygen species (ROS). Initially, during cellular respiration in mitochondria, electrons pass through protein complexes in the electron transport chain: when an electron leaks or is prematurely transferred to molecular oxygen, a superoxide anion (O_2_^−^) is created. Ionizing radiation, such as X-rays and gamma rays, can also ionize water molecules to produce hydroxyl radicals. Moreover, enzymatic reactions can inadvertently generate ROS, as can the Fenton Reaction (which can be simplified as Fe^2+^ + H_2_O_2_ → Fe^3+^ + OH^−^ + HO^•^), where iron (Fe^2+^) and hydrogen peroxide react to form hydroxyl radical (HO^•^), and hydroxyl ion (OH^−^) which is not a radical. Peroxisomes, which play roles in metabolism and detoxification, produce ROS and rely on catalase to break down hydrogen peroxide into water and oxygen. This process can lead to an excess of ROS if catalase is overwhelmed. Additionally, the enzyme NOX produces superoxide ions by transferring electrons from NADPH to oxygen. Endoplasmic Reticulum (ER) stress and the Unfolded Protein Response also contribute to ROS production. Indeed, the ER, responsible for protein synthesis, folding, and modification, can become stressed by an overload of misfolded proteins due to factors like nutrient deprivation, hypoxia, and calcium level changes. This stress can induce oxidative stress, while the cell’s response, aimed at restoring ER homeostasis, increases mitochondrial activity, and thereby ROS production. Lastly, xenobiotics subjected to metabolization and microbes, either during their metabolism or as a defense mechanism, can generate ROS.

Given their strong ability to react chemically with their environment, ROS are well known to lead to several pathological consequences [34]. Surprisingly, depending on the specific context, ROS can exhibit both protective and deleterious effects on the organism, with the same molecule exerting opposing effects on biological processes [35]. The impact of these effects relies on the interplay between the production of ROS and their clearance rates, wherein antioxidants and scavengers assume a critical role [36]. When this balance is disrupted, these reactive molecules can interact unopposed with, and damage, vital cellular components such as DNA, proteins, and lipids. This molecular damage can compromise the normal functions of cells, leading to tissue injury. This harmful state, characterized by the overwhelming presence of oxidative species and the resultant cellular and tissue damage, is termed “oxidative stress” [34].

### 4.1. Generation Pathways of Reactive Oxygen Species

As previously mentioned, ROS originate from O_2_, a diradical, meaning it has two unpaired electrons in its molecular orbital. Despite being relatively stable under standard conditions, its diradical nature makes O_2_ more reactive than if it were a closed-shell molecule, and prone to participate in various chemical reactions, especially those involving electron transfer. 

When O_2_ acquires an electron, primarily from the mitochondrial electron transport chain as earlier described, it gives rise to superoxide anion (O_2_^−^). This species carries both a negative electric charge and an unpaired electron, making it both an anion and a radical. It can be seen as the precursor to ROS and it also plays a role in the formation of RNS. It is a free radical, i.e., a molecule that contains one free unpaired electron in its outer orbital. However, O_2_^−^ does not seem to act as a potent oxidizing agent despite its name. Instead, its detrimental effects stem from its secondary interactions, particularly when it reacts with nitric oxide (NO). Due to its low permeability, the actions of superoxide are primarily related to the compartment in which it is produced; however, it can also influence the surrounding cellular environment by crossing the cell’s ion channels, exerting its effects at a distance from the production point, even with a half-life, at normal pH, of about 5 s [37]. 

Hydrogen peroxide (H_2_O_2_) is generated from O_2_^−^ through a reaction called dismutation (i.e., a reaction in which the same species is both reduced and oxidized), in which two superoxide molecules are transformed in H_2_O_2_ and O_2_, catalyzed by the enzyme superoxide dismutase (SOD). H_2_O_2_ is not a free radical, and therefore it does not directly oxidize other molecules. However, it can move quickly through the cell membranes and, transforming into a potent compound (hydroxyl radical, described further in the text), it can exert the effects of ROS at different sites instead of its production site. Indeed, due to a non-enzymatic reaction that requires iron in its reduced state (Fe^2+^), known as the Fenton reaction, the most reactive compound known, hydroxyl radical (HO^•^), can be generated from H_2_O_2_. The OH^•^ can also be formed through the reaction between O_2_^−^ and nitric oxide (NO), and the Haber–Weiss reaction (which can be simplified as O_2_^−^ + H_2_O_2_ → HO^•^ + OH^−^ + O_2_) [13,38]. Due to its tendency to react quickly with the first available molecule, this radical has a short half-life [36]. To prevent the continuation of the reaction chain, catalases can decompose H_2_O_2_ into water and O_2_, thereby preventing the formation of HO^•^. Because of this cooperative capability, both SOD and catalase are classified as antioxidants. 

The main ROS described and their formation mechanisms are schematically depicted in Figure 2.

### 4.2. The Effects of ROS on Cerebral Vascular Tone

In the brain, whose metabolism relies almost exclusively on the presence of oxygen, ROS appear to have a decisive influence on cerebral flow [13,39]. Indeed, the brain has a unique neurovascular coupling structure, facilitating functional and anatomical interactions among neurons, astrocytes, pericytes, microglia, and blood vessels, known as the neurovascular unit [40]. This unit allows the selective adaptation of blood flow in different brain regions, influenced by ROS and NO, which act on smooth muscle cells and pericytes [13,41]. For example, in animal models, ROS have been shown to have a role in microvascular reperfusion disturbances [25]. 

ROS can affect cerebral vascular smooth muscle tone in several biochemical pathways [42], potentially in a more pronounced way in the presence of cerebrovascular diseases [43]. The overall vasoactive effect varies based on (1) the specific type of molecule involved and (2) its concentration. For example, O_2_^−^ can act as a direct endothelium-dependent vasodilator and a smooth muscle vasoconstrictor through the inactivation of NO [44,45]. Indeed, O_2_^−^, as well as H_2_O_2_, can affect the vascular tone, causing vasodilation, by activating calcium-dependent potassium channels (K_ca_), which are the most common K^+^ channel expressed on cerebral arteriolar musculature, and which are sensitive to the redox state of the environment [13]. By contrast, O_2_^−^ leads to vasoconstriction through a powerful inhibition of endothelial NO-mediated dilation. The equilibrium between these two opposite results depends on the net concentration of the reactive species. Indeed, findings from numerous in vitro and in vivo animal studies confirm that both O_2_^−^ and H_2_O_2_ exhibit a biphasic impact on cerebral blood vessels, contingent upon their concentration. At low concentrations, O_2_^−^ leads to vasodilation, while at high concentrations, it causes vasoconstriction. Similarly, H_2_O_2_, when present in very high concentrations, may initially induce vasoconstriction, and later lead to vasodilation [44].

In addition to the aforementioned mechanisms, the modulation of cerebral arteriolar tone through chloride channels has also been suggested [46]. Moreover, O_2_^−^ can react with arachidonic acid and other unsaturated fatty acids to form isoprostanes. These are known as strong vasoconstrictors that can cause a reduction in cerebral blood flow. Thus, O_2_^−^ can indirectly further affect vascular tone, and may be of pathological interest in traumatic brain injuries [13,44]. Finally, H_2_O_2_ would also act in some cases as a membrane depolarizer, exerting, with hyperpolarization, a vasodilatory effect [13,44,45]. 

Lastly, the production rates of ROS and antioxidants vary according to different oxygenation levels, each following unique kinetic patterns [35,47]. Therefore, it is conceivable to assume that the vasoactive activity of ROS, as well as RNS and antioxidants, may depend not only on the types of ROS and their concentration, but also on a third feature, i.e., the kinetics of their production and degradation. Indeed, compensatory mechanisms might come into play when a stimulus enters a new steady state. This can occur especially in pathological conditions, such as in acute brain injury or in critically ill patients, where elevated ROS levels may increase acutely, and persist for an extended period. 

The vascular impacts of ROS on cerebrovascular tone are diverse and involve complex mechanisms. These also include interactions with NO and RNS, which will be further discussed later.

### 4.3. NADPH Oxidases

The nicotinamide adenine dinucleotide (phosphate) oxidases (NADH/NADPH oxidases, or Nox) are prominently involved in the enzymatic generation of ROS [48]. These membrane-bound enzymes produce superoxide radicals by utilizing NADH or NADPH as electron donors. In mammals, there exist seven isoforms of Nox. Among them, Nox1, Nox2, Nox4, and Nox5 are expressed in various tissues, such as the endothelium, vascular smooth muscle cells, fibroblasts, and perivascular adipocytes [49]. Under normal physiological conditions, vascular Nox exhibits a relatively low baseline activity. However, in response to various stimuli, like cytokines, the enzyme activity can be augmented both in the short term and over an extended period. This response to stimuli disrupts vascular homeostasis, leading to pathological changes and associated health issues, such as increased blood pressure [50].

The involvement of NADH and NADPH reinforces the notion that the redox state is critical in influencing the tone of cerebral circulation. Indeed, interestingly, studies have shown that NADPH oxidase activity occurs at a significantly higher magnitude in cerebral arteries than in systemic arteries [12]. As electron acceptors, both NADH and NADPH facilitate the activity of Nox enzymes, leading to the production of O_2_^−^ [26,39], which is, in the end, responsible for the impact on the vascular tone. Thus, ROS production derived from NADPH oxidase may contribute both to maintaining normal physiological vascular tone and to vascular pathology.

### 4.4. ROS as Signaling Molecules in HIF-1α Regulation

ROS have more effects than expected, and they could work as signal molecules. In this way, it is thought that ROS play a role in the degradation of Hypoxia Inducible Factor HIF-1α, as there is evidence of a negative correlation between HIF-1α and ROS [41]. HIF-1α is a cytosolic transcription factor that moves to the nucleus during hypoxia. It binds to HIF-β and activates the expression of genes containing hypoxia-responsive elements (HRE), producing hypoxia-related proteins such as erythropoietin (EPO). The production of ROS is implicated in the proposed “normobaric oxygen paradox” [51], whereby intermittent hyperoxic stimulation could paradoxically mimic a hypoxic-like condition, leading to the activation of HIF-1α [52]. Moreover, ROS production via NADPH oxidase, especially Nox2, may intervene in the evoked HIF-1α synthesis and stability [53]. These mechanisms may be especially relevant in brain cells, as they tend to increase ROS production under injured conditions. However, the existence of this phenomenon and its potential effects on brain tissue have not yet been thoroughly investigated.

### 4.5. Regional Variability in Oxidative Stress Responses within Brain Vasculature

It has been observed that the mechanisms that intervene in oxidative equilibrium may show differences between the contexts of cerebral and systemic circulation [12,54]. Indeed, differences may be seen in susceptibility to oxidative stress among the cerebral microvessels in various brain regions [55]. Additionally, neurons situated in particular areas of the brain, such as the hippocampus and the cerebellar granule cell layer, are more prone to oxidative stress [56]. These regional differences may also be influenced by the local availability of NO during brain injuries, which has been demonstrated in experimental animal models [27]. Therefore, it appears conceivable that even small changes in ROS and RNS concentrations can lead to clinical changes via their effect on the brain vasculature, especially during brain injuries and cerebrovascular diseases. Nevertheless, further human research is required to gain data and a deeper comprehension of these aspects, moving beyond theoretical discourses.

## 5. Reactive Nitrogen Species 

Reactive Nitrogen Species (RNS) are a group of highly reactive nitrogen-containing molecules that are generated as by-products of nitrogen metabolism. These molecules are crucial in various physiological and pathological processes, including vascular control [57]. 

### 5.1. Generation Pathways of the Main Reactive Nitrogen Species

Nitric oxide (NO) is the most well-known and significant member of the RNS family. Also formerly known as endothelium-derived relaxing factor (EDRF), it is constantly generated in the body through the enzymatic action of nitric oxide synthases (NOS) on L-arginine and oxygen. There are three distinct isoforms of NOS present in the body: iNOS (inducible), eNOS (endothelial), and nNOS (neuronal) [58,59]. NOS generates in loco NO, which readily exerts its biological activity. Although its half-life is limited to a few seconds, NO can react with hemoglobin, forming S-nitroso hemoglobin, or be converted to nitrite. S-nitroso hemoglobin (SNO-Hb) is a derivative of hemoglobin modified by adding a NO group to a cysteine thiol group, in a reaction called S-nitrosylation. 

Nitrite (NO_2_^−^) is a chemical compound that consists of one nitrogen atom and two oxygen atoms. It is an anion that can form salts such as sodium nitrite (NaNO_2_) or potassium nitrite (KNO_2_). SNO-Hb and NO_2_^−^ salts act as NO reservoirs, which can be transported through systemic circulation and released under certain conditions to promote vasodilation and increase blood flow to tissues distant from the site of origin [60,61]. 

Other molecules can be categorized into RNS, produced as metabolites in NO decomposition or by reactions between other compounds such as CO_2_ and ROS. For example, peroxynitrite (ONOO^−^) is formed from the reaction between NO and superoxide ions, thus reducing NO concentrations [62]. Although evidence suggests that NOS is responsible for nitroxyl (HNO) production, it is still unclear how this compound is synthesized in vivo [63,64]. Indeed, HNO can also be generated after the reduction in NO levels by mitochondrial cytochrome C, xanthine oxidase, ubiquinol, hemoglobin, and manganese superoxide dismutase (SOD) [64].

Nitrate (NO_3_^−^) is a chemical compound that consists of one nitrogen atom and three oxygen atoms. Similarly to NO_2_^−^, it carries a negative charge and is thus found in salt chemical form. A significant amount of nitrate comes from leafy green vegetables. Salivary glands actively take up nitrate from circulation, resulting in a saliva concentration 10-fold higher than in plasma [65]. A certain quantity of NO_3_^−^ is converted to NO_2_^−^ by reductase enzymes in the salivary glands [65]. Finally, some nitrites are absorbed when ingested and become part of the “nitrate–nitrite–nitric oxide pathway” [66]. Kapil et al. have demonstrated how nitrate supplementation can lower blood pressure, and that depleting oral microbiomes can jeopardize endogenous nitrate production, leading to hypertension [67]. There are other endogenous ways to synthesize nitrate and nitrite [66]. Over the years, many trials have established the conversion of NO to nitrite as occurring naturally in vivo. This reaction has been linked to hypoxia, deoxyhemoglobin, and lower pH [68,69,70,71,72]. Furthermore, a significant increase in plasma nitrite was displayed during the administration of 80 ppm iNO in a murine model [73]. Given all these observations, nitrates and nitrites can be considered a NO reservoir [74]. Indeed, while NO has a half-life limited to 2 ms, that of nitrite is much longer, and can quickly spread all over the organism [66].

The formation of the main RNS described is schematically depicted in Figure 3.

### 5.2. The Effects of RNS on Cerebral Vascular Tone

#### 5.2.1. Nitric Oxide 

Since its recognition as an endothelium-derived relaxing factor [75,76], several studies have described NO as a strong vasodilator agent [58]. After being synthesized in the endothelium, NO increases the levels of cyclic guanosine monophosphate (cGMP) in smooth muscles by stimulating soluble guanylyl cyclase. cGMP finally inhibits the proteins responsible for contraction, causing vasorelaxation [58,77]. In adjunct to this mechanism, NO induces a reduction in intracellular calcium, mediated by ATPases that actively transport calcium inside the sarcoplasmic/endoplasmic reticulum [58,77]. Other pathways toward vasodilation include the reduction in 20-hydroxyeicosatetraenoic acid [78] and prostaglandins [79]. NO may be key in regulating cerebral blood flow (CBF) [80]. Indeed, in hyperoxia [81,82,83], hypoxia [81,82], and hypercapnia [84], it was found to be an essential mediator of CBF in animal models. Moreover, NO also plays a leading role in coupling CBF and neuronal activity [85]. 

Interestingly, CO_2_ may act as a regulator of NO. Indeed, the process of adjusting cerebral vascular tone in response to changes in arterial carbon dioxide partial pressure (PaCO_2_) is known as chemoregulation, which could be the main trigger of endothelial NO-release [86]. In humans, hypocapnia leads to vasoconstriction and reduced CBF, while hypercapnia and changes in perivascular pH result in vasodilation and increased CBF [86]. Cerebral endothelial cells and astrocytes have been shown to release NO under normocapnic conditions, while NO production increases during hypercapnia and decreases during hypocapnia, regardless of pH levels [87]. It has been proposed that these NO variations in response to PaCO_2_ are specific to NOS regulation, and that administering exogenous NO may influence the CO_2_-dependent chemoregulation mechanism [87].

Several studies have explored the role of NO in critical neurologic illnesses. Indeed, early stages of severe brain injuries recognize a depletion of NO, probably contributing to secondary brain damage [88]. By contrast, when present, NO reduces injuries and has neuroprotective properties in ischemia/reperfusion animal models, both in relation to stroke and cardiac arrest [88]. For this reason, a prospective trial has been performed, proving the feasibility of iNO administration in cardiac arrest, and showing promising results in terms of survival [89]. In subarachnoid hemorrhage, NOS inactivation is associated with alterations in microvascular density and hemostasis, leading to rebleeding, intracranial hypertension, and larger hemorrhage volume [90]. Conversely, NO reduction is also correlated with a lower grade of neuroinflammation and better neurobehavioral function in rats [91]. Studies in humans have showed a correlation between elevated asymmetric dimethylarginine (which is an inhibitor of iNOS, and thus of the vasodilator molecule NO) and vasospasm and worse outcomes [92]. Another small human study found elevated levels of NO metabolites, especially in subjects with poorer outcomes [93]. Although the results may be controversial, the literature suggests deleterious effects of NO depletion, supported by increased microthrombi formation and reduced cortical activity [88]. A pilot human trial established the safety of iNO administration and showed promising results in treating delayed cerebral ischemia [94].

Pathways in traumatic brain injury (TBI) appear more complex. Mechanical insults may induce the upregulation of iNOS and the over-production of NO and ONOO^−^. The RNS stimulate the production of glutamate, which triggers nNOS. Nevertheless, NO plays an uncertain role in this disease, and its implications are still unknown [95]. Studies in newborn animal models are also contradictory. Some reports showed protective properties of NO related to vasodilation, reducing ROS concentrations and scavenging radicals; others linked it to deleterious effects and greater brain injuries. These discrepancies might depend on the production timing and NO concentrations [96]. The brain controls blood flow by interacting with vessel diameter. Autoregulation ensures constant blood supply; changes in systemic blood pressure induce vasodilation or vasoconstriction. Additionally, the brain increases blood flow in areas where neurons are more active through a mechanism named neurovascular coupling. Although other pathways and elements participate, NO plays a significant role in both of them. NOS inhibition (and thus the lowering of NO production) increases the lower limit of mean arterial blood pressure where autoregulation acts, reducing its efficiency. Through neurovascular coupling, blood flow is directed to more active areas, where glutamate (the main excitatory neurotransmitter) induces calcium entry in neurons and the activation of neuronal nNOS, finally resulting in NO production and vasodilation [79].

Finally, as a NO reservoir, the vasodilating effects of nitrites have long been investigated. Although some studies failed to find any vasoactive activity [97], most studies confirmed that nitrite has vasorelaxant effects through the activation of guanylate cyclase [60,98,99,100,101].

In summary, NO is a potent vasodilator, playing diverse roles in regulating blood flow and brain protection. Its mechanism is intricate and involves numerous interactions. When it comes to brain injuries, a deficiency in NO could contribute to secondary brain damage, while NO possesses neuroprotective qualities in animal models of ischemia/reperfusion, including stroke. Interestingly, NO inhalation has been suggested as a neuroprotective intervention during cardiopulmonary resuscitation [102]. Consequently, extensive research is needed to better understand its mechanisms and potential therapeutic effects.

#### 5.2.2. Peroxynitrite

Peroxynitrite (ONOO^−^) has shown vasoactive properties, producing vasorelaxation in arterioles in vitro [28] and in vivo [36], although its potency seems very low, and is up to 50-fold less than that of NO [103]. The effect does not appear to involve the endothelium [104], and the vasoactive mechanism remains unclear. Some authors have proposed that the effect is linked to the opening of ATP-sensitive potassium channels [36,105], directly activating the channels or reducing ATP concentration by interfering with cellular metabolism [36,105,106]. Other mechanisms proposed to explain this vasodilating activity include the elevation of cGMP levels, membrane hyperpolarization via K^+^ channel activation, the activation of myosin phosphatase activity, and interference with cytosolic calcium movement and cellular membrane Ca^2+^ entry [28]. 

However, in contrast with all the above results, Daneva et al. induced an increase in ONOO^−^ by upregulating iNOS in mice, reporting an increase in pulmonary arterial pressure by impeding calcium entry in smooth muscle cells of pulmonary arteries [107]. Similarly, Ottolini et al. investigated the same pathway in diet-induced obese mice, finding a link between obese hypertension and high ONOO^−^ levels [108]. Finally, in vitro studies [29,30] have observed that peroxynitrite may induce vasoconstriction in cerebral arteries. This effect is likely due to the inhibition of the cerebral K^+^-dependent calcium channel. Interestingly, the addition of glutathione inhibited this cerebral vasoconstrictive effect.

Therefore, the net effects of peroxynitrite appear conflicting, and are not completely understood, but are probably dose- and time-dependent [109].

#### 5.2.3. Nitroxyl 

HNO has been proven to act as an endothelium-derived vasorelaxant [110,111,112,113] that is not susceptible to tolerance [111]. Moreover, HNO has multiple effects, including inhibiting platelet aggregation, limiting vascular smooth muscle proliferation, and interacting with metallo and thiol-containing proteins.

Although we failed to find in the literature any research focused on the impact of this molecule on cerebral tone, we have reported its effects on the general vasculature. Indeed, HNO promotes vasodilation by activating several molecular pathways [64,112,113,114]. These pathways include the stimulation of soluble guanylyl cyclase (sGC), leading to an increase in cGMP. From animal studies, it has been inferred that the sGC-cGMP pathway may be necessary and sufficient for HNO-induced vasodilation in vivo [115]. Others proposed that HNO may activate ATP-sensitive potassium (K_ATP_) channels and voltage-dependent potassium (K_v_) channels, resulting in the outflow of potassium ions. This causes the cell membrane to become hyperpolarized and leads to a reduction in intracellular calcium levels. Finally, following an elegant animal study, Eberhardt et al. suggested HNO-mediated vasodilation via the transient receptor potential channel A1 (TRPA1) and calcitonin gene-related peptide (CGRP) [116]. This HNO–TRPA1–CGRP signaling pathway could be a crucial component in the neuroendocrine regulation of vascular tone, mediated by HNO.

Figure 4 presents a visual abstract illustrating how ROS and RNS might influence the cerebrovascular tone.

## 6. Antioxidants

Antioxidants are substances that protect from the deleterious effects of free radicals. Their action, which can be enzymatic and non-enzymatic, can be further divided into two groups: (1) preventing the generation of ROS and blocking the generated ROS, and (2) repairing the damages the free radicals have caused. 

Reducing the level of antioxidants leads to an increase in oxidative stress, thus indirectly influencing the cerebrovascular tone. Indeed, a more oxidative cellular environment can contribute to an increase in vascular tone by reducing the activity of NO [117,118]. For this reason, antioxidant mechanisms are widely expressed in vascular cells and brain tissue.

### 6.1. Superoxide Dismutase

Superoxide dismutase (SOD) is an enzymatic defense system catalyzing O_2_^−^ into H_2_O_2_. It exists in three different forms: copper–zinc SOD (SOD1 or Cu-Zn SODI), manganese SOD (SOD2 or MnSOD), and extracellular SOD (SOD3 or ecSOD). These are some of the most important protective mechanisms against oxidative stress in the body, and decreased SOD levels may be associated with a worse outcome in acute ischemic stroke (AIS) patients, faster vascular damage, and blood–brain barrier breakdown [119]. SOD has been proposed as a marker of cardiovascular alterations in hypertensive and diabetic patients, as fluctuations in serum levels are linked to alterations in vascular structure and function [120].

SOD1 is predominantly found in the cytoplasm and nucleus of cells, where it helps neutralize superoxide radicals generated during normal metabolic processes. The copper and zinc ions are essential to its enzymatic activity. SOD2 is usually located in the mitochondria, thus representing a critical defense mechanism against oxidative damage to mitochondrial components. Finally, SOD3 is primarily found in the extracellular matrix and fluids, acting as a defense outside the cells, protecting from oxidative stress caused by superoxide radicals released from inflammatory cells. Moreover, it is active in the arterial wall, strategically between the endothelium and smooth muscle, thus controlling reactivity to oxygen levels and vascular tone [83] by regulating nitric oxide. Indeed, from animal studies, we can infer that SOD regulates in vivo both the basal tone and the vascular response to different pressures of oxygen [83]. In TBI patients, SOD levels may be reduced already during the first day, and may remain low for 7 days [121]. Indeed, exogenous SOD has been proposed as a therapeutic adjunct in several conditions [122], and it may reduce vasospasm after SAH [123]. The reduction in cerebral vasospasm in experimental SAH animal models obtained by administering SOD [31] underscores the significant involvement of superoxide anions and redox equilibrium in vascular regulation, particularly under pathological conditions.

### 6.2. Glutathione Redox System

*Glutathione peroxidase* (GPx) is an enzyme that reduces hydrogen peroxide to water via a tripeptide called glutathione (GSH). GSH is the most common thiol, i.e., a molecule that contains a sulfhydryl group (SH), and is present in the cytosol and its organelles. Composed of glutamine, cysteine, and glycine, glutathione works as an electron donor, passing from its reduced state (GSH), the active form, to an oxidized state (GSSG). The enzyme glutathione reductase converts GSSG back into GSH, using NADPH as an electron donor. GPx plays a central role in reducing hydrogen peroxide to water, keeping it away from the production of hydroxyl radicals. Moreover, GSH can directly and independently neutralize free radicals, acting as a non-enzymatic antioxidant. Indeed, it can donate electrons to free radicals, an activity called “scavenging”, oxidizing them to form GSSG. 

This mechanism is one of the most important antioxidant processes in the body. Indeed, GSH is the most abundant intracellular antioxidant, as its concentration within the cytoplasm where it is produced is in the mmol range, where the ratio GSH:GSSG is normally maintained around 100:1 (but it can fall lower than 10:1 during oxidative stress). It can move into the extracellular fluids, but plasma levels are 1000 times less concentrated than intracellular levels. An exception is in the lining fluid in the lungs, where the concentration is 140 times higher than in the plasma. 

Other molecules can also intervene in this system. For example, as already described, NADPH serves to keep GSH reduced. The pentose phosphate pathway generates NADPH, and glucose-6-phosphate dehydrogenase (G6PD) is one of the enzymes in the pathway. Deficiency of this enzyme, which affects millions of people worldwide, can result in GSH deficiency, predisposing one to oxidative injury. Thiamine is a cofactor inhibiting the pentose phosphate. Indeed, evidence has been found that thiamine administration may alleviate oxidative stress, even in cases where it is not attributable to thiamine deficiency [124,125]. Finally, selenium is necessary to glutathione peroxidase activity. Indeed, this enzyme contains selenium in the active site [126]. In sepsis and septic shock, selenium is abnormally low and, when infused, has been shown to reduce mortality [127]. This has also been reported by a recent meta-analysis, even if with a low quality of evidence [128].

Glutathione deficiency is a common finding in neurological disorders [129], and it may be related to several cardiovascular diseases [130]. Even in the brain vasculature, GSH may have protective effects. Despite apparently being the most convenient and safe method of GSH administration, oral GSH is not frequently utilized in clinical trials due to its limited effectiveness [131]. 

An alternative approach to increasing GSH production involves employing N-acetylcysteine (NAC). NAC is smaller than GSH, can move into the cells, and acts as a cysteine precursor, a crucial factor in limiting GSH synthesis [127]. NAC administration presents various neuroprotective effects, such as a reduction in the size of cerebral stroke [132] and better outcomes in neurological deficits and disability [133]. Moreover, NAC can potentially reduce ischemia–reperfusion injury [134,135], and may enhance the vasodilator effects of other compounds, such as acetylcholine [136]. As shown with other antioxidants, NAC treatment may effectively reduce cerebral vasospasm in SAH animal models [32], potentially carrying significant clinical implications that will require clinical trials to determine efficacy, despite promising pre-clinical data.

### 6.3. Vitamin C (Ascorbic Acid)

Vitamin C, also known as ascorbic acid (AA), is a crucial water-soluble antioxidant that plays multiple roles in the body, including acting as a cofactor for various enzymes and inhibiting the generation of ROS. Additionally, it directly scavenges ROS and RNS while repairing other oxidized scavengers. One fascinating aspect of ascorbic acid is its neuroprotective activity, and AA has been proposed as a therapeutical adjunct in several neurological conditions [137]. 

Indeed, the brain exhibits one of the highest ascorbic acid concentrations in the body, thanks to an active uptake from the bloodstream. The brain concentration of AA varies from 200–400 μM in the extracellular space to 10 mM intracellular, i.e., almost 100 times the concentration in the plasma [138,139]. 

The effect of AA on vessels started to be investigated more than forty years ago [140]. It has demonstrated that it has the potential to enhance impaired endothelium-dependent vasodilation in peripheral arteries in various conditions characterized by endothelial dysfunction [141]. However, the impacts of vitamin C on peripheral and cerebral circulations may differ [142]. 

Although blood hyperoxia does not appear to be convincingly associated with cerebral vasospasm following a subarachnoid hemorrhage [143], animal studies have shown that intracisternal injected Oxy-Hb (oxyhemoglobin) may result in a concentration-dependent contraction of cerebral artery strips. However, the presence of ascorbic acid significantly modifies the biological activity of Oxy-Hb, suppressing its vasoconstrictor activity and minimizing its ability to reduce vasodilator actions [144].

The cerebrovascular effects of AA were recently evaluated in an elegant study by Mattos et al. [33], investigating the effects of isocapnic hyperoxia (IH) on CBF and metabolism. Ten male participants underwent a 10 min IH trial with intravenous saline and AA infusion (3 g). AA infusion inhibited ROS production and preserved NO bioavailability, as indicated by the reduced ROS biomarkers and unchanged nitrite levels. The infusion also prevented regional and total CBF decline, and restored the cerebral metabolic rate of oxygen (CMRO_2_) during IH. This study further confirms that ROS play a role in CBF regulation and metabolism, and antioxidants such as AA can have indirect vasoactive effects.

Considering all this, the use of ascorbic acid in protecting the cerebral vascular tone during high oxidative stress conditions, like those seen in critically ill patients or those with acute brain injuries, requires further study, as there are a few adverse reactions [145].

### 6.4. Vitamin E (Tocopherols and Tocotrienols)

Vitamin E is not a single molecule, but rather a group of fat-soluble substances with similar vitamin activity, which includes four tocopherols (alpha, beta, gamma, delta) and four tocotrienols (alpha, beta, gamma, delta). The most biologically active isoform among them is alpha-tocopherol [146]. These compounds protect the integrity of cellular membranes by shielding them from the free radicals that can generate and by directly neutralizing superoxide and hydroxyl radicals. Vitamin E may have potential neuroprotective effects, particularly in neurodegenerative diseases, stemming from its activity in reducing neuroinflammation [147].

Besides its role as an antioxidant, vitamin E plays a significant role in various cellular processes, such as in the vascular vessels. For example, in patients with coronary spastic angina, administering alpha-tocopherol acetate at 300 mg/day restored flow-dependent vasodilation [148]. This improvement was accompanied by a reduction in levels of plasma lipid peroxidation substances and a decrease in anginal attacks. 

Endothelial cells, when replete with vitamin E, are better equipped to prevent blood cell components from adhering to their interior surface in blood vessels. Moreover, vitamin E enhances the expression of two enzymes that suppress arachidonic acid metabolism, leading to the increased release of prostacyclin from the endothelium, promoting blood vessel dilation and inhibiting platelet aggregation [6]. Its administration can improve arteriolar compliance [149] and endothelium-dependent relaxation [150]. Furthermore, a meta-analysis has indicated that vitamin E might provide protection in preventing ischemic stroke [151], but another recent one failed to show a benefit [152], suggesting that further well-designed randomized controlled trials are required to reach a conclusive result.

### 6.5. Other Antioxidants

Other antioxidants can be used as promising adjunct therapies in critically ill patients or those with brain injuries. For instance, catalase, which degrades hydrogen peroxide when administered intracisternally in meningitis models, caused a reduction in the elevation of rCBF (regional cerebral blood flow), ICP (intracranial pressure), and brain water content, primarily caused by superoxide or its derivatives [153]. Indeed, hydrogen peroxide induced relaxation in cerebral arteries, regardless of the presence of intact endothelial cells. However, these relaxations were suppressed by catalase, while SOD (superoxide dismutase) had no impact, suggesting that the relaxations resulted from the direct effect of hydrogen peroxide [154]. Likewise, other than improving endothelial function [155], flavonoids may elicit favorable impacts on the vascular system, resulting in ameliorating cerebrovascular blood flow [156], potentially preventing vasospasm in SAH [157], and with therapeutic effects on ischemic stroke-induced models [158].

Vitamin D has demonstrated the potential to improve neurological outcomes in brain-injured patients with significant vitamin D deficiency [159]. Interestingly, its levels have shown a correlation with intracranial aneurysm rupture in patients with SAH [160]. Furthermore, it has been suggested that the vasoprotective effects induced by vitamin D may be attributed to a reduction in oxidative stress [161]. Nevertheless, its ability to effectively prevent oxidative stress remains inconclusive, and its role as an antioxidant cannot currently be definitively established [162].

Vitamin A may have both antioxidant and oxidant activity [163]. It is a fat-soluble vitamin that exists in two primary forms: retinol, a directly absorbable form found in animal products, and carotenoids, which the body can convert into Vitamin A. Its use in TBI patients has been shown to have potential neuroprotective effects, reducing brain lesion sizes [164]. Although it seems to play a key role in the development of cerebral vasculature [165], we failed to find specific studies evaluating its effects on cerebrovascular tone.

Finally, circulating uric acid may act as a significant water-soluble antioxidant, particularly against peroxynitrites in hydrophilic settings. However, within the cells, uric acid may turn into a potent pro-oxidant, triggering oxidative stress cells and mitochondria and promoting the production of inflammatory cytokines [35], in a paradoxical dualism [166]. This exemplifies the difficulties in evaluating these data and underscores the need for high-quality clinical studies. 

## 7. Discussion 

The regulation of vascular tone in cerebral vessels is a complex process that involves multiple factors and mechanisms [167], which include pressure-induced regulation, shear stress, local metabolism, and the control of vascular diameter by neuronal activity. As previously described, the presence of ROS, RNS, and antioxidants can result in vasodilatory and vasoconstrictive effects in cerebral blood vessels, with a complex interplay activity [168]. The main effects of the various oxidative and nitrosative species involved in modulating the cerebral vascular tone are reported in Table 2. 

In situations of intense hyperoxia, where there is a significant rise in O_2_^−^, the existing NO can be neutralized by O_2_^−^. The rate of this reaction matches the combined rates of other known superoxide degradation reactions [169]. The chemistry of NO suggests that its half-life is likely controlled by ROS, which increases in proportion to the partial pressure of oxygen in the brain, leading to an increase in arterial tone. Interestingly, during hyperbaric oxygenation in animal models, the vasodilating effect of superoxide dismutase when animals were pre-treated with a NO-synthase inhibitor was not seen [170], suggesting that one cause of vasoconstriction in the brain during hyperoxia might be the neutralization of NO by O_2_^−^, impacting its natural ability to relax blood vessels.

Besides playing a role in regulating the tone of cerebral vessels under physiological conditions, the equilibrium between ROS, RNS, and antioxidants becomes crucial under pathological conditions. Indeed, the brain is especially susceptible to oxidative injury due to its rapid metabolic rate and proneness to ischemic damage. Oxidative stress occurs due to an imbalance between the generation and detoxification of free radicals, increasing reactive species [34]. This equilibrium is disrupted in several acute brain injuries, including TBI and stroke [121,171,172], and chronic neurodegenerative diseases [173]. For instance, studies have demonstrated that SOD activity decreases 24 h following a TBI and continues to be reduced 7 days after a severe TBI [121]. Similarly, there appears to be a disruption in cerebrovascular tone, involving mechanisms that may encompass both nitrogen [174] and oxygen-reactive species [175]. The SOD activity also resulted in reductions in the serum of stroke patients for several days [176]. Therefore, the injured brain may suffer secondary damage resulting not only from the well-known direct oxidative activities of ROS, but also from the lesser-explored impacts these molecules have on blood vessels.

In these patients, the balance of redox reactions becomes essential, making antioxidants potential therapeutic agents to counteract oxidative damage. Antioxidants have been considered as a potential adjunctive treatment for various critical illnesses and have notably been suggested as a potential therapeutic option for patients with sepsis [177]. Nevertheless, they have been shown to be ineffective in a clinical study of sepsis [178]. Antioxidants can neutralize ROS and RNS, prevent their generation, and repair the damage caused by free radicals. For this reason, they have been proposed and tested to reduce the effects of reactive species on cerebral blood flow [33]. Indeed, even the mildest reduction in the regional cerebral blood may lead to cognitive dysfunction [179]. Therefore, utilizing antioxidants may potentially serve as a relatively safe supplementary therapy for specific critically ill patients or those with brain injuries. This may mitigate cognitive impairments in the former group and reduce secondary brain damage in the latter. For example, ascorbic acid has demonstrated its ability to preserve NO bioavailability, prevent CBF decline, and protect the cerebral metabolic rate of oxygen during conditions of high oxidative stress [33]. NAC administration has shown neuroprotective effects in reducing the size of cerebral stroke and improving neurological deficits [133]. In the context of patient-tailored therapy, it is plausible that antioxidant therapy should also be targeted toward those who truly require it, and it is conceivable that antioxidant levels should be evaluated in the intensive care unit, as most ICU patients experienced swift and severe deficiencies in antioxidants [180]. 

While antioxidants hold promise for offering potential benefits to brain injuries and the control of cerebrovascular tone, there is currently limited human evidence available. To our knowledge, as of now, there are no approved antioxidant therapies for TBI and stroke. Thus, further research is required to elucidate their precise mechanisms of action, effects on the vascular tone, optimal dosages, combinations of antioxidants, and potential interactions with other treatment modalities. 

## 8. Conclusions 

ROS, RNS and antioxidants contribute to regulate the tone of the cerebral vessels in different ways. Understanding the complex interplay between ROS, RNS, antioxidants and their impacts on cerebral vascular tone is essential for deepening our comprehension of brain physiology and mechanisms of secondary damage in brain injuries. This knowledge holds the potential to pave the way for the development of effective therapeutic approaches to neurological disorders, with antioxidants emerging as promising supplementary treatments. These treatments aim to counteract the detrimental effects of oxidative stress and maintain cerebral blood flow, particularly in individuals with acute brain injuries or those in critical conditions. As the field of study progresses, it may reveal novel avenues to enhance outcomes and the quality of life for individuals suffering from neurological injuries.

## Figures and Tables

**Figure 1 ijms-25-03007-f001:**
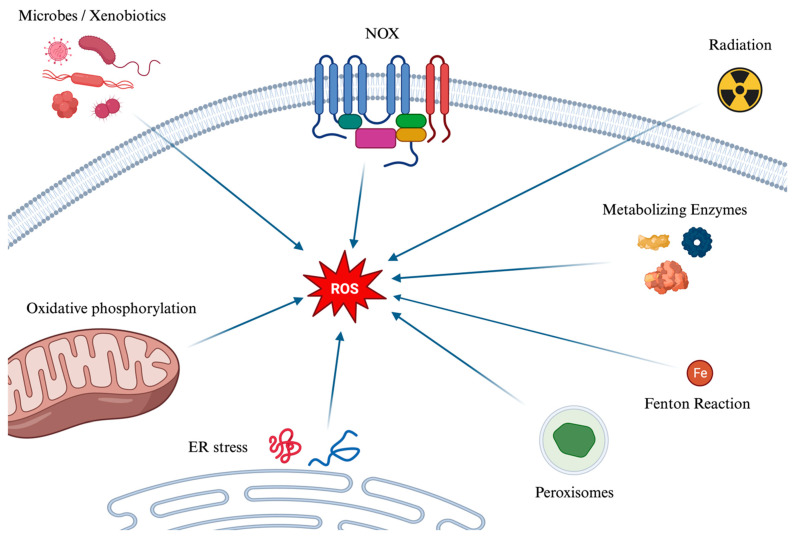
A graphical summary of the main pathways of the formation of ROS. Reactive oxygen species (ROS) are generated through various mechanisms, impacting cellular functions positively and negatively. They originate from microbes and xenobiotics metabolism, NOX enzyme activity, ionizing radiation effects, and metabolic enzymes, leading to superoxide, hydroxyl radicals, and other ROS forms. The Fenton reaction, peroxisomes’ metabolic activities, and the breakdown of fatty acids further contribute to ROS accumulation. Endoplasmic reticulum stress and the unfolded protein response enhance oxidative stress by increasing mitochondrial activity and oxidative phosphorylation, where electron leakage can form superoxide. ROS: Reactive Oxygen Species. NOX: NADPH oxidase. ER: Endoplasmic Reticulum.

**Figure 2 ijms-25-03007-f002:**
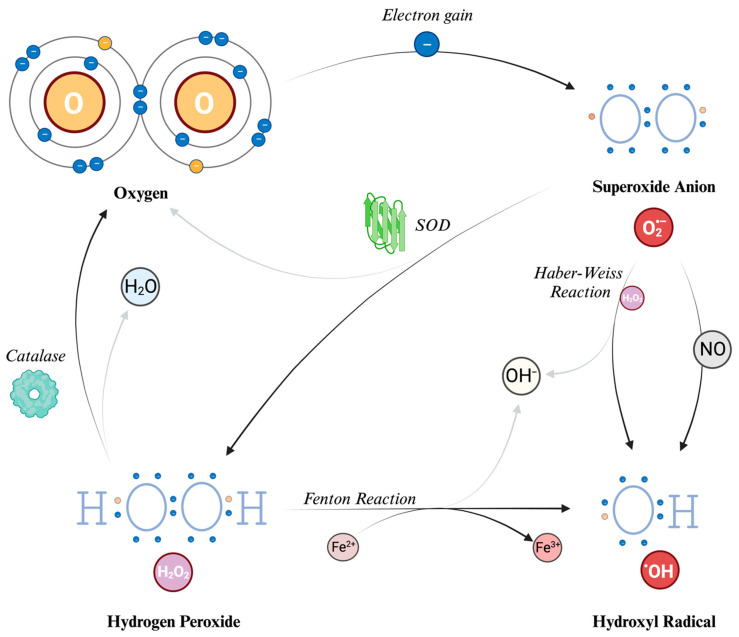
Reactive Oxygen Species (ROS): their structure and formation. The process begins with molecular oxygen (O_2_), which has two unpaired electrons in the outer orbital. Thus, it can acquire an electron to form the superoxide anion (O_2_^−^), which is a radical (i.e., with an unpaired electron). Superoxide dismutase (SOD) converts superoxide into hydrogen peroxide (H_2_O_2_) and oxygen. Catalases can further break down H_2_O_2_ into water and oxygen. The Haber–Weiss reaction generates hydroxyl radicals (HO^•^) and hydroxyl ions (OH^−^) from superoxide and H_2_O_2_. The Fenton reaction also produces hydroxyl radicals and hydroxyl ions by reducing H_2_O_2_ in the presence of iron ions. O: oxygen atom. H: hydrogen atom. H_2_O: water molecule. O_2_^−^: superoxide anion. H_2_O_2_: hydrogen peroxide. HO^•^: hydroxyl radical. OH^−^: hydroxyl ion. NO: nitric oxide. Fe^2+^: iron in reduced state. Fe^3+^: iron in oxidized state. SOD: superoxide dismutase.

**Figure 3 ijms-25-03007-f003:**
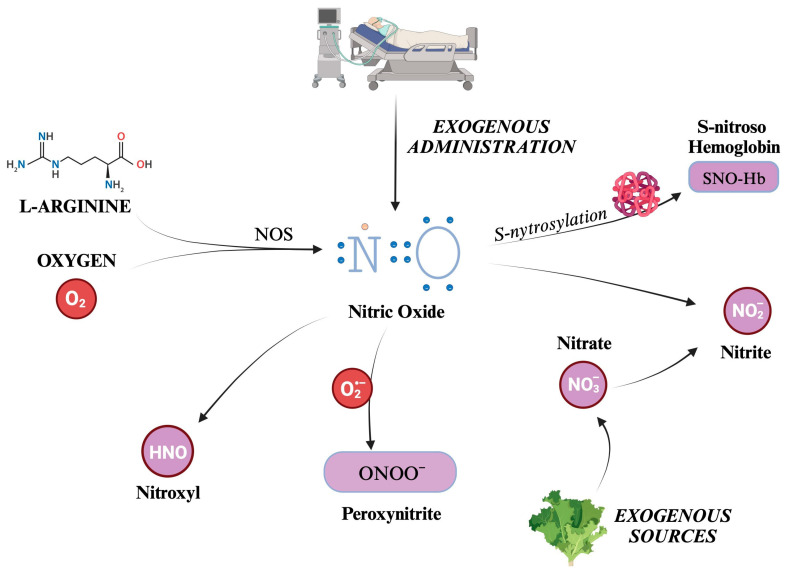
Overview of the main reactive nitrogen species (RNS) and their formation pathways. The figure illustrates the generation of nitric oxide (NO), a small, gaseous molecule that acts as a signaling molecule in the body. It is produced by the enzyme nitric oxide synthase (NOS) from the amino acid L-arginine and molecular oxygen. S-nitrosylation adds a NO group to the thiol group of a cysteine residue, for example, Hb. Otherwise, NO can be transformed into salts or esters of nitrous acid, containing the anion nitrite (NO_2_^−^), which can be converted back to NO under certain conditions. Nitrates are taken into the body from external sources, such as food or supplements. They can be converted into nitrites (and thus potentially into NO). This process is known as the nitrate–nitrite–NO pathway and is an alternative source of NO, especially when the enzymatic production of NO is impaired. Peroxynitrite is a highly reactive nitrogen species formed by NO with O^2−^. Finally, nitroxyl (HNO) is an isoelectronic reactive nitrogen species with NO, formed by the reduction of NO or the decomposition of certain NO donors. O_2_: molecular oxygen. NO: nitric oxide. NOS: nitric oxide synthase. N: nitrogen atom. Hb: hemoglobin. NO_2_^−^: nitrite. NO_3_^−^: nitrate. ONOO^−^: peroxynitrite. HNO: nitroxyl.

**Figure 4 ijms-25-03007-f004:**
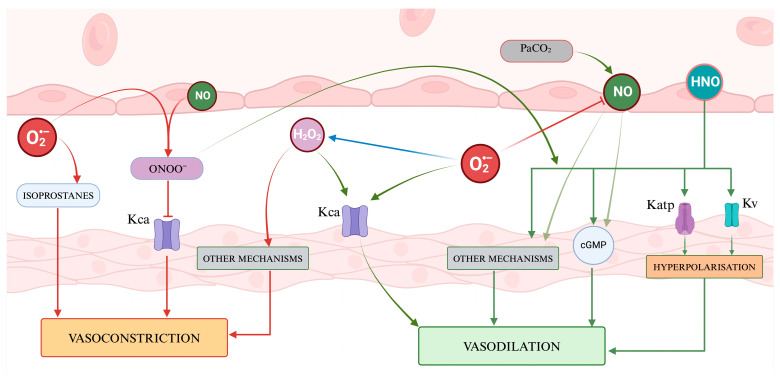
A representation of the regulation of vascular tone by Reactive Oxygen Species and nitric species. ROS presents a dual role, with hydrogen peroxide acting as a signaling molecule inducing vasodilation through the activation of potassium channels and vasoconstriction through other mechanisms, while superoxide contributes to vasodilation by affecting K_ca_ channels and vasoconstriction by generating isoprostanes. Nitric oxide and the other RNS mainly promote vasodilation via cGMP and several mechanisms. The balance of vasoactive molecules and their net effect on vascular smooth muscle contribute to determining the cerebral vascular tone.

**Table 1 ijms-25-03007-t001:** Experimental models and techniques for studying oxidative stress in cerebral vasculature. Overview of the various experimental approaches employed or that could be utilized to investigate the impacts of oxidative stress on cerebral vasculature. Each method offers a different point of view of the mechanisms by which ROS, RNS, and antioxidants may affect cerebral vascular tone and blood flow.

Model/Technique	Investigations on Cerebral Vasculature
Animal models (e.g., rats, rabbits, etc.) [25,26,27]	ROS/RNS effects on cerebral blood flow, microperfusion and autoregulation
In vitro studies [28,29,30]	Molecular mechanisms modulating the endothelial and smooth muscle cell response
Use of antioxidants [31,32,33]	Therapeutic neuroprotective therapies and modulation of vascular tone in oxidative stress conditions
Diagnostic tools	Non-invasive measurement of the effects of ROS/RNS on cerebral blood flow and metabolism
Pharmacological interventions	Investigation of the causal relationship among ROS, RNS, antioxidants, and vascular tone

**Table 2 ijms-25-03007-t002:** Mechanisms of oxidative stress-mediated modulation of cerebral vascular tone. This table outlines the various oxidative and nitrosative species involved in modulating cerebral vascular tone, and their mechanisms and effects (either vasodilation or vasoconstriction).

Mechanisms	ROS/RNS Involved	Effect on Cerebral Vascular Tone
Interaction with ion channels	O_2_^−^, H_2_O_2_, HNO, ONOO^−^	Modulation of K^+^ channels, leading to vasodilation or vasoconstriction [13,36,105,116]
Interaction with nitric oxide	O_2_^−^	Inactivation of NO, leading to vasoconstriction [44,45]
Other endothelial and myogenic mechanisms	O_2_^−^, H_2_O_2_, NO, ONOO^−^	Vasodilation/vasoconstriction [28,46,60]

## Data Availability

Not applicable.

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
