# Peer review of "Oxidative Stress and Cerebral Vascular Tone: The Role of Reactive Oxygen and Nitrogen Species"

_ijms, 2024, doi:10.3390/ijms25053007_

Round 1
Reviewer 1 Report
Comments and Suggestions for Authors
The topic of this review is good. But, this article needs lots of improvements.
1. The title of this review article must be changed to represent the of the work. Mere listing of terms such as ROS, RNS, etc. will not provide any meaning at all.
2. There is no need for Table 1, as this information can be easily described in a paragraph.
3. In the end of introduction, state how this review is different than previously published ones.
4. Figure 1 caption must be compressed to just 3-4 lines. All the detailed descriptions must be explained in the main text.
5. Section heading 2.1 does not provide any meaning. What is the meaning of main ROS? Please revise it.
6. Section 2.4 and 2.5 headings must be revised to provide actual meaning of the paragraph.
7. Same issue with the section heading 3.1 similar to 2.1.
8. Figure 4 caption is exhaustive, reduce it.
9. Is Figure 4 necessary? I do not understand its use.
10. In section heading 4.3 and 4.4, write their original name too.
11. Combine the conclusion and future perspectives.
12. Draw a graphical abstract.
13. For section 4, reproduce at least 3-4 study results figures from research papers.
14. Explain briefly about Cerebral Vascular Tone in a separate section. May be section 2 suits it.
15. Why only 4 antioxidants have been described in section 4? Add 3 more sub-sections for other antioxidants.
Comments on the Quality of English LanguageMinor editing is required.
Author Response
The topic of this review is good. But, this article needs lots of improvements.
R0. Thank you for conducting a thorough review of our paper. We have carried out a comprehensive examination of the manuscript and have provided detailed responses to each of your comments.
- The title of this review article must be changed to represent the of the work. Mere listing of terms such as ROS, RNS, etc. will not provide any meaning at all.
R1. We changed the title to: "Oxidative Stress and Cerebral Vascular Tone: The Role of Reactive Oxygen and Nitrogen Species”
We believe this title aligns more closely with the paper's objective, offering readers a clearer understanding of its contents.
- There is no need for Table 1, as this information can be easily described in a paragraph.
R2. In accordance with your recommendation, we have eliminated Table 1. We determined that creating a new section was unnecessary, as most of the factors were already covered in the text. This would have led to repetition. Instead, we have integrated its contents into the paragraph.
- In the end of introduction, state how this review is different than previously published ones.
R3. Thank you for highlighting this aspect. We failed to find literature reviews that are both comprehensive and accessible to a broad audience. For this reason, our goal is to present a paper that elucidates the complex mechanisms of oxidative stress in a clear manner, while also being informative for a diverse readership. This includes providing the latest results present in the literature. To clarify this, accordingly to your comment, we have added a statement at the end of the introduction.
- Figure 1 caption must be compressed to just 3-4 lines. All the detailed descriptions must be explained in the main text.
R4. The caption has been condensed, and comprehensive explanations for the details have been provided within the main text. Thank you for bringing this to our attention.
- Section heading 2.1 does not provide any meaning. What is the meaning of main ROS? Please revise it.
R5. Your point is valid. We have updated the heading to “Generation Pathways of Reactive Oxygen Species”.
- Section 2.4 and 2.5 headings must be revised to provide actual meaning of the paragraph.
R6. The headings have been updated.
- Same issue with the section heading 3.1 similar to 2.1.
R7. Acknowledged. We have revised the heading to “Generation Pathways of Reactive Nitrogen Species”.
- Figure 4 caption is exhaustive, reduce it.
R8. The caption has been erased along with the Figure.
- Is Figure 4 necessary? I do not understand its use.
R9. Thank you for your input. We erased Figure 4.
- In section heading 4.3 and 4.4, write their original name too.
R10. We have included the original names as requested.
- Combine the conclusion and future perspectives.
R11. We have expanded the discussion with additional text, so we believe the conclusion section should remain separated.
- Draw a graphical abstract.
R12. A graphical abstract has been drawn.
- For section 4, reproduce at least 3-4 study results figures from research papers.
R13. In Section 5 (previous section 4), a new figure has been introduced, substituting the previous one.
- Explain briefly about Cerebral Vascular Tone in a separate section. May be section 2 suits it.
R14. A new section about Cerebral Vascular Tone has been added.
- Why only 4 antioxidants have been described in section 4? Add 3 more sub-sections for other antioxidants.
R15. Different antioxidants like Vitamin A have been included, based on your recommendations, in “other antioxidants” section.
Reviewer 2 Report
Comments and Suggestions for Authors
Thank you for submitting your manuscript for consideration at IJMS
The manuscript covers a very important topic which is the effect of ROS, and RNS on cerebral vascular tone. The author talked about how ROS and RNS are generated as well as the role of antioxidants in their neutralization however the author missed two important issues in this review
- ROS are formed in the brain at a high rate since they are crucial signaling molecules in the brain for induction of synaptic plasticity and memory formation
-Under physiological conditions, ROS act as essential signaling molecules, necessary for the proper formation of learning and memory processes. However, during aging, ischemia, trauma, or neurodegenerative diseases, the levels of ROS increase to levels higher than the antioxidant machinery of the cells can handle, and therefore their beneficial signaling role becomes outweighed by the ambiance of oxidative damage that they create and focusing the review only on the deleterious effect of the species is somewhat biased
- the authors missed vitamin A (retinol or beta carotene) as a potential antioxidant in thier review
Comments on the Quality of English Language
- English need to be revised as there are some typos
Author Response
Thank you for submitting your manuscript for consideration at IJMS. The manuscript covers a very important topic which is the effect of ROS, and RNS on cerebral vascular tone. The author talked about how ROS and RNS are generated as well as the role of antioxidants in their neutralization however the author missed two important issues in this review.
R0. We appreciate the reviewer's kind words. We concur that the paper addresses a topic that is not only of significant importance but also of potentially great interest to a wide audience. We have responded to each comment in a point-by-point manner.
- ROS are formed in the brain at a high rate since they are crucial signaling molecules in the brain for induction of synaptic plasticity and memory formation. Under physiological conditions, ROS act as essential signaling molecules, necessary for the proper formation of learning and memory processes. However, during aging, ischemia, trauma, or neurodegenerative diseases, the levels of ROS increase to levels higher than the antioxidant machinery of the cells can handle, and therefore their beneficial signaling role becomes outweighed by the ambiance of oxidative damage that they create and focusing the review only on the deleterious effect of the species is somewhat biased.
R1. We have included a paragraph in the discussion that discusses this aspect, which is instrumental in providing a more comprehensive understanding of the overall topic.
- The authors missed vitamin A (retinol or beta carotene) as a potential antioxidant in thier review.
R2. We have included a discussion about the potential antioxidant properties of vitamin A, as per your recommendation.
Reviewer 3 Report
Comments and Suggestions for Authors
Major Comments:
1. Title looks not matched with the content.
2. Abbreviations are not denoted properly.
3. No specific problem statement in introduction section.
4. I did not find any attractive correlation. Need to redesign the contents.
5. Figures 1, 2, 3, and 4: Not interesting for general readers. These are typical figures.
6. Cerebral Vascular Tone: No correlations found.
7. Role of ROS and RNS in Cerebral Vascular Tone Regulation: need in details.
8. Imbalance and Oxidative Stress in Cerebral Vascular Diseases: need in details.
8. Experimental Approaches and Methodologies: need in details.
9. Need more attractive figures.
10. Conclusion: not properly written.
Comments on the Quality of English LanguageExtensive revisions
Author Response
R0. We thank the reviewer for the helpful and constructive comments. We believe they have contributed to enhancing the quality of our paper.
- Title looks not matched with the content.
R1. We have revised the title to more accurately reflect the content of the paper.
- Abbreviations are not denoted properly.
R2. All the abbreviations have been checked and corrected accordingly, a list of the abbreviations and acronyms have been inserted at the end of the paper.
- No specific problem statement in introduction section.
R3. At the end of the introduction a statement has been added, accordingly.
- I did not find any attractive correlation. Need to redesign the contents.
R4. The contents have been redesign.
- Figures 1, 2, 3, and 4: Not interesting for general readers. These are typical figures.
R5. We agree they are typical figures, but at the same time they are necessary to understand aspects which are not clear for every general readers. Previous Figure 4 has been erased
- Cerebral Vascular Tone: No correlations found.
R6. We apologize for any previous confusion, and we believe that the revisions made in the reviewed manuscript have enhanced its clarity. Also, a new section has been introduced.
- Role of ROS and RNS in Cerebral Vascular Tone Regulation: need in details.
R7. A final graphical abstract has been reported and concepts have been clarified in the manuscript.
- Imbalance and Oxidative Stress in Cerebral Vascular Diseases: need in details.
R8. Details about this aspect have been added in the discussion section.
- Experimental Approaches and Methodologies: need in details.
R9. Details regarding the Experimental Approaches and Methodologies have been incorporated following your recommendations.
- Need more attractive figures.
R10. One figure has been erased, and a new one has been created
- Conclusion: not properly written.
R11. The conclusion has been rephrased.
Round 2
Reviewer 2 Report
Comments and Suggestions for Authors
Thank you for the revised version much better now
Comments on the Quality of English LanguageMuch better
Author Response
Thank you for your interest in our paper; your contributions have significantly improved it.
Reviewer 3 Report
Comments and Suggestions for Authors
Though corrections are not properly done, I recommend adding two comprehensive tables:
Table: Mechanisms of Oxidative Stress-Mediated Modulation of Cerebral Vascular Tone
Table: Experimental Models and Techniques for Studying Oxidative Stress in Cerebral Vasculature
Comments on the Quality of English LanguageThough corrections are not properly done, I recommend adding two comprehensive tables:
Table: Mechanisms of Oxidative Stress-Mediated Modulation of Cerebral Vascular Tone
Table: Experimental Models and Techniques for Studying Oxidative Stress in Cerebral Vasculature
Author Response
We thank you for these additional suggestions; the two tables have been added as per your recommendation.
Round 3
Reviewer 3 Report
Comments and Suggestions for Authors
Now it can be accepted. Thank you.
Comments on the Quality of English Language-
Author Response
Thank you for your consideration.